# Fe_2_O_3_/Porous Carbon Composite Derived from Oily Sludge Waste as an Advanced Anode Material for Supercapacitor Application

**DOI:** 10.3390/nano12213819

**Published:** 2022-10-28

**Authors:** Shubing Tian, Baoling Zhang, Dong Han, Zhiqiang Gong, Xiaoyu Li

**Affiliations:** 1College of Mechanical and Electronic Engineering, Shandong University of Science and Technology, Qingdao 266590, China; 2College of New Energy, China University of Petroleum (East China), Qingdao 266580, China; 3State Grid Shandong Electric Power Research Institute, Jinan 250003, China

**Keywords:** Fe_2_O_3_, porous carbon, hydrothermal treatment, supercapacitor, oily sludge

## Abstract

It is urgent to improve the electrochemical performance of anode for supercapacitors. Herein, we successfully prepare Fe_2_O_3_/porous carbon composite materials (FPC) through hydrothermal strategies by using oily sludge waste. The hierarchical porous carbon (HPC) substrate and fine loading of Fe_2_O_3_ nanorods are all important for the electrochemical performance. The HPC substrate could not only promote the surface capacitance effect but also improve the utilization efficiency of Fe_2_O_3_ to enhance the pseudo-capacitance. The smaller and uniform Fe_2_O_3_ loading is also beneficial to optimize the pore structure of the electrode and enlarge the interface for faradaic reactions. The as-prepared FPC shows a high specific capacitance of 465 F g^−1^ at 0.5 A g^−1^, good rate capability of 66.5% retention at 20 A g^−1^, and long cycling stability of 88.4% retention at 5 A g^−1^ after 4000 cycles. In addition, an asymmetric supercapacitor device (ASC) constructed with FPC as the anode and MnO_2_/porous carbon composite (MPC) as the cathode shows an excellent power density of 72.3 W h kg^−1^ at the corresponding power density of 500 W kg^−1^ with long-term cycling stability. Owing to the outstanding electrochemical characteristics and cycling performance, the associated materials’ design concept from oily sludge waste has large potential in energy storage applications and environmental protection.

## 1. Introduction

Supercapacitors (SCs) were treated as a type of energy storage device because of their advantages of outstanding power density, superior cycling stability, fast charging/discharging, and high efficiency compared with traditional capacitors and batteries [1,2,3]. It has a broad application prospect in the fields of consumer electronics, hybrid-electric vehicles, emergency power supply systems, smart distributed grid systems, and so on. However, low energy density is the most significant obstacle limiting the practical application of SCs [4,5].

To promote energy density, tremendous methods have been developed according to the relationship of E = CV^2^/2. Considering the quadratic relationship between energy density (E) and operating voltage (V), enlarging voltage should be an efficient way to increase energy density [6]. Many studies have shown that the organic electrolytes would provide a high operating voltage up to 3.0 V, whereas the high cost and environmental toxicity hinder their applications [7]. Another way to expand operating voltage is to assemble an asymmetric supercapacitor device, which possesses two different electrode materials used as the cathode and anode, respectively. According to the recent reports, aqueous-based ASC with different cathodes and anodes could extend the voltage to 1.6–2.2 V [8,9]. Thus, ASC could be a promising approach to enhance energy density. In addition, a direct strategy to improve the energy density (E) is exploring electrode materials with higher specific capacitance (C) according to the above relationship [10]. As for ASC, it is equally important to increase the specific capacitance of either cathode or anode for enhancing the energy density. However, the development is uneven between the cathode and anode. Numerous research efforts have been devoted to high-performance cathode materials [11,12]; there are only a few reports of novel anode materials.

Due to the huge surface area, massive electrical conductivity, high mechanical power, and less density, traditionally activated carbon (or modified version with a higher porosity, such as carbon nanotubes or graphene) has been commonly employed as anode materials for supercapacitors [13,14,15]. However, these carbonaceous anodes often exhibit small specific capacitances because the charge storage only relies on the surface accumulation of electrolyte ions. Some pseudo-capacitive anodes based on reversible redox reactions received extensive concern to meet the charge balance of the cathode in an ASC device, for example, some transition metal oxides of Fe_2_O_3_ [16], Fe_3_O_4_ [17], Co_3_O_4_ [18], and the binary metal oxides of MgMoO_4_ [19], NiMoO_4_ [20], CoMoO_4_ [21], NiCo_2_O_4_ [22], et al. Recently, Fe_2_O_3_ was intensively treated as SCs’ anode due to the large theoretical specific capacitance (3625 F g^−1^), natural abundance, low cost, and environmental friendliness [23]. However, as to the few reported Fe_2_O_3_-based electrodes, their electrochemical performances, such as low specific capacitance and poor cycling life, are far from satisfactory because of the shortcomings of the poor conductivity and limited specific surface area. An emerging new concept to overcome these shorts is to introduce Fe_2_O_3_ nanostructures onto the conductive substrates to form composites [24,25,26,27]. It is feasible to use carbon materials as substrates to form a carbon/Fe_2_O_3_ composite. Tang et al. [28] reported an anode of Fe_2_O_3_ nanowires grown on carbon fiber paper, which possesses a high specific capacitance. Yue et al. [29] synthesized the Fe_2_O_3_/CNTs composite with shell-core architecture with the microwave-assisted method. The CNTs substrate significantly improved the conductivity of the composite material, and the capacitance performance was enhanced compared to the pure Fe_2_O_3_ electrode material [30,31]. However, the electrochemical performance of these composites might be affected because of poor contact [32,33,34,35]. It is necessary to study a new type of Fe_2_O_3_/carbon composite material with a stable nanostructure that could fully exploit the advantages of both components and enhance the electrochemical performance effectively [36].

In this work, the Fe_2_O_3_/HPC composite was synthesized through a hydrothermal route from oily sludge (OS) waste. The α-Fe_2_O_3_ nanorods were deposited on the surface of the HPC substrate without destroying the porous structure. The advantages of both HPC substrate and active Fe_2_O_3_ nanorods could be fully exploited, and the capacitance performance of the composite could be effectively enhanced. The HPC substrate was beneficial to the dispersion of nanoparticles and increased the interfaces between electrode and electrolyte. The appropriate loading of Fe_2_O_3_ nanorods was helpful to maintain the aperture characteristic of the composite and improve the diffusion ability of electrolyte ions in the electrode material. When used as the anode for SCs, the obtained FPC materials could deliver a competitive specific capacitance (465 F g^−1^ at 0.5 A g^−1^), an excellent rate capability, and cycling stability.

## 2. Materials and Methods

### 2.1. FPC Preparation

The OS sample used in this study was oil tank bottom sludge which was collected during the tank cleaning process in a petroleum storage depot of SINOPEC. The OS sample was filtered and dried at 105 °C for 24 h. The proximate and elemental analysis of the OS precursor is shown in Appendix A. All the other chemical reagents were of AR grade and used without further purification.

The preparation process of FPCs is illustrated in Figure 1. In the beginning, OS-based hierarchical porous carbon (HPC) substrates were synthesized according to previous work [37]. Briefly, oily sludge was carbonized at 700 °C under N_2_ atmosphere. After being washed by HF, the carbon residue was activated by KOH at a mass ratio of 1:3 under 700 °C. In the typical synthesis of the FPC composite, 100 mg of HPC was dispersed and sonicated in distilled water, and 864 mg of FeCl_3_·6H_2_O and 284 mg of Na_2_SO_4_ were dissolved into the suspension. Na_2_SO_4_ here was employed as a structure-directing agent to facilitate the relatively uniform growth of nanorods. The resulting mixture was transferred into a Teflon-lined stainless-steel autoclave (100 mL) and hydrothermally treated in an oven at 90 °C for 12 h. Following air cooling to ambient temperature, the samples were collected and washed with distilled water to remove the adsorbed ions. Then, the FeOOH/HPC composite was obtained after drying at 80 °C. Further, the brown Fe_2_O_3_/HPC composite named FPC-90 could be collected by calcining FeOOH/HPC under Ar atmosphere at 400 °C for 2 h. As a comparison, the product synthesized under a hydrothermal temperature of 180 °C was obtained as FPC-180.

Moreover, the pure Fe_2_O_3_ electrode was prepared under similar conditions as mentioned above. A piece of titanium (Ti) foil cleaned with distilled water, ethanol, and acetone successively was treated as a substrate and placed in the autoclave during the hydrothermal treatment [38]. All the other experimental conditions were in accordance with FPC-90.

### 2.2. Characterization

A field emission scanning electron microscope (FESEM, JSM-7600F, JEOL, Tokyo, Japan) was used for the evaluation of FPCs’ samples’ morphology. The transmission electron microscope (TEM, JEM-2100UHR, JEOL, Tokyo, Japan) was used to characterize the microstructures of the FPCs. To investigate the proportion of FPCs, a thermogravimetric analysis (TGA, TGA/SDTA851, Mettler Toledo, Zurich, Switzerland) was carried out from room temperature to 700 °C at a rate of 10 °C/min under air flow. An X-ray diffractometer (XRD, PANalytical X’Pert Pro, Panalytical, Almelo, The Netherlands) was used to analyze the crystal structures of FPCs, and an X-ray photoelectron spectroscopy (XPS, ESCALAB MK II, Thermo Scientific, Waltham, MA, USA) analysis was conducted to detect the surface chemical state of the samples. The specific surface areas (S_BET_) and pore size distributions were calculated from the N_2_ adsorption-desorption isotherms obtained on the automatic analyzer at 77 K (Micromeritics ASAP 2020, Micromeritics, Norcross, GA, USA).

### 2.3. Electrochemical Measurements

The electrochemical performance of the as-obtained FPCs composites was tested on a CHI 660E electrochemical workstation (Shanghai Chenhua Science Technology Corp., Ltd., Shanghai, China) in 1 M Na_2_SO_4_ aqueous electrolyte. Cyclic voltammetry (CV), galvanostatic charge/discharge (GCD), and electrochemical impedance spectroscopy (EIS) were carried out to characterize the electrochemical properties of the working electrode. In a typical three-electrode system, a platinum plate and an Ag/AgCl electrode were employed as counter and reference electrodes, respectively. For a working electrode, the powder of FPC samples was mixed with carbon black and polyvinylidene-fluoride with the ratio of 8:1:1 in N-methyl pyrrolidone to form a slurry. Then, the slurry was painted on nickel foam (1 cm × 1 cm) and used as working electrodes. The mass loading of FPC on the electrode was about 3.0 mg cm^−2^.

The specific capacitance *C* (F g^−1^) was calculated using the following equation,
(1)$$C=\frac{I\;∆t}{m\;∆V}$$
where *I*, ∆t, *m*, and ∆V are the discharge current (A), discharge time (s), active mass (g) on working electrode, and the potential window (*V*), respectively.

Further, an ASC device was fabricated to further discuss the electrode properties of the composites. An MnO_2_ nanowire/porous carbon composite (MPC) [39] was employed as the cathode and the FPC composite was used as the anode. The cathode and anode were separated by polypropylene film and immersed in 1 M Na_2_SO_4_ solution to form a two-electrode system. The mass ratio of the cathode and anode in the ASC device was calculated by the charge balance of both electrodes. The specific capacitance was calculated using the above equation, where *m* means the total mass of both cathode and anode. Moreover, the energy density (*E*) and power density (*P*) of the ASC device can be calculated by the following equations, respectively.
(2)$$E=\frac{CV^{2}}{2\times 3.6}\;$$
(3)$$P=3600\times \frac{E}{∆t}$$
where *C*, V, and ∆t are separately the specific capacitance, voltage, and discharge time.

## 3. Results and Discussion

### 3.1. Characterizations of Samples

Our strategy is to deposit Fe_2_O_3_ nanoparticles onto the porous carbon structure and maintain the networks of the carbon substrate. The as-received FPC-90 exhibits an irregular distribution of Fe_2_O_3_ nanorods on the surface of HPC as shown in the SEM image in Figure 2b. These nanorods are uniform in size: about 20 nm in diameter, and 100 nm in length. The heterogeneous nucleation could be taken to explain the growth of the nanorods [40]. Commonly, heterogeneous nucleation will be promoted and more favorable due to the lower activation energy barrier. Therefore, the nucleation on the surface of the carbon substrate requires a lower saturation concentration than that in the solution. It means that the reaction is more likely to occur on the surface of HPC.

In addition, the Fe_2_O_3_ nanorods on FPC-90 grow vertically on the substrate rather than along the HPC surface. This is due to the epitaxial crystal growth that would go along with the easy direction of crystallization when the nucleation was limited by the deposition conditions. The high concentration of the precursor solution facilitates vertical crystalline growth on the substrate. As a comparison, the Fe_2_O_3_ nanoparticles on FPC-180 (Figure 2c) synthesized at a hydrothermal temperature of 180 °C show an accumulation of grains with irregular sizes, which is different from FPC-90. This is because that higher temperature could overcome the surface energy barrier between the nanorods and enable them to agglomerate into bundles. Therefore, the appropriate hydrothermal temperature plays a significant role in determining the structures of Fe_2_O_3_ nanorods, which is critical to the electrochemical properties of the composite. As a control sample which is shown in Figure 2a, the pure Fe_2_O_3_ shows the similar size and nanorods’ morphology with FPC-90, while the nanorods on the surface of FPC-90 are arranged more sparsely. This is because the porous structure of carbon substrates provides a rougher surface, which results in the disordered growth of the nanorods.

The TEM image in Figure 2d gives insights into the micromorphology of the FPC-90, where nanorods with diameters of around 20 nm are deposited on the surface of the carbon substrate. Furthermore, the co-existence of C, Fe, and O elements is proven by the results of the corresponding elemental mapping images (Figure 2e–h).

More structure characteristics of the samples are investigated by XRD as shown in Figure 3a. The XRD pattern of the synthesized Fe_2_O_3_ shows distinct peaks which can be well assigned to α-Fe_2_O_3_ (JCPDS No. 33-0664) [29]. The HPC substrate is amorphous carbon which does not have any sharp characteristic peaks. It has broad peaks at around 2*θ* = 26° and 44°, which are superimposed with the Fe_2_O_3_ sharp peaks in the patterns of FPC-90 and FPC-180. The sharp shape of the diffraction peaks reveals the samples possess high crystallinity. Based on these XRD patterns, the crystallite size of the Fe_2_O_3_ nanoparticles is determined using Scherrer’s relation [41]. The average size of Fe_2_O_3_ in the sample of α-Fe_2_O_3_, FPC-90, and FPC-180 turns out to be about 13, 19, and 32 nm, respectively, consistent with the SEM and TEM results.

Figure 3b gives the TGA curves of different samples. Compared with HPC, the initial oxidation temperature of FPC-90 and FPC-180 are increased close to 350 °C, because the Fe_2_O_3_ deposited on the surface of FPCs hinders the oxidation of the carbon substrate. In addition, the weight loss for HPC, FPC-90, and FPC-180 is 95.4%, 61.6%, and 31.4%, respectively. Considering that Fe_2_O_3_ is stable when heated in air, the loading capacity of Fe_2_O_3_ in FPC-90 and FPC-180 is calculated as 35.4% and 67.1%, respectively, indicating that the Fe_2_O_3_ content increases with increasing hydrothermal temperature.

XPS is used to investigate the surface element composition of FPC. As shown in Figure 3c, the full survey spectra are dominated by the signals of C, O, and Fe elements in the composites. In high-resolution XPS spectra of the Fe2p region (Appendix A), two distinct characteristic peaks at 711.6 and 725.1 eV are observed and correspond to a spin-orbit couple of Fe 2p_3/2_ and Fe 2p_1/2_, respectively [42]. In addition, these two peaks are accompanied by another two satellite peaks situated at 719.9 and 733.5 eV, which are consistent with the characterization of Fe^3+^.

The N_2_ adsorption–desorption results are illustrated in Figure 3d to deeply investigate the porous structure of FPCs. The FPC-90 sample exhibits a typical combined I/IV type isotherm. The steep increase in adsorption capacity at low relative pressure is related to the micropores, while the easing increase at higher relative pressure and the hysteresis loop is caused by the capillary condensation of N_2_ in the mesopores [43]. However, the FPC-180 sample exhibits type I isotherm, which means the dominance of micropores. The isotherm of α-Fe_2_O_3_ could be classified as type IV with nearly no adsorption capacity at low relative pressure, indicating that there are no micropores in the prepared α-Fe_2_O_3_ nanorods. The calculated BET surface area and pore size distributions based on these isotherms are listed in Appendix A. The specific surface area of FPC-90 is 851.3 m^2^ g^−1^ with a total pore volume of 0.739 cm^3^ g^−1^. It means that the porous structure of the HPC substrate is not significantly decreased by the deposition of Fe_2_O_3_ nanorods. The porous structure of FPC-90 is beneficial to its capacitive performance as an electrode material. With the increasing hydrothermal temperature, the formed Fe_2_O_3_ particles gradually block the pore structure in FPC-180, which possesses a surface area of 635.2 m^2^ g^−1^ and a total pore volume of 0.382 cm^3^ g^−1^. Then, the pure α-Fe_2_O_3_ only exhibits a surface area of 168.6 m^2^ g^−1^ with slight micropore distribution, in the absence of the contribution of the HPC substrate.

### 3.2. Capacitance Performance

To evaluate the supercapacitor performance of the as-prepared FPC electrodes, electrochemical tests are performed in a three-electrode configuration. The GCD curves are recorded at different current densities from 0.5 A g^−1^ to 20 A g^−1^ (Appendix A) in the potential window of [−1–0 V]. The nearly mirror-image current response indicates good reversibility of these electrodes. The slight distortion of the triangle shape manifests the combination of EDLC caused by the interface adsorption between the electrode material and electrolyte and pseudo-capacitive behavior of Fe_2_O_3_, which arises from a reversible reaction [23]:(4)$${\mathrm{Fe}}_{2}O_{3}{+2\mathrm{Na}}^{+}{+2e}^{-}↔{\mathrm{Na}}_{2}{\mathrm{Fe}}_{2}O_{3}$$

The comparison of different samples at 2 A g^−1^ are shown in Figure 4a. It is clear that the FPC-90 electrode exhibits a longer discharging time than FPC-180 and α-Fe_2_O_3,_ which manifests higher specific capacitance. Despite the higher content of Fe_2_O_3_ in FPC-180, the specific capacitance is decreased. On the one hand, the larger Fe_2_O_3_ particle size would reduce the interface at which the pseudo-capacitive behavior occurs. On the other hand, the excessive Fe_2_O_3_ nanoparticles on the surface could also weaken the porous characteristics of the composites, hindering the rapid diffusion of the electrolyte ions and the adsorption effects on the surface of the electrode materials, thus reducing the EDLC performance. The synthesized pure α-Fe_2_O_3_ nanorods possess the lowest specific capacitance with respect to the composites. Although it is an effective pseudo-capacitive material with higher theoretical specific capacitance, its smaller specific surface area and poorer pore structure could not provide a sufficient electrode–electrolyte reaction interface. It follows that during the hydrothermal reactions, the HPC substrate provides more dispersed deposition sites for Fe^3+^ ions loading, thus avoiding the agglomeration of the dense FeOOH structures. The dispersed Fe_2_O_3_ nanorods in the composites obtained by a further burning reaction facilitate the transport of electrolyte ions, thereby improving the capacitance of the material.

The rate performance of these different electrodes calculated from GCD curves is illustrated in Figure 4b. The highest specific capacitance is tested as 465 F g^−1^ at a current density of 0.5 A g^−1^ for the electrode of FPC-90, which is much better than those Fe_2_O_3_-carbon composite electrode materials reported recently [34,35,44] (more details are shown in Appendix A). With the increase of the current density, the specific capacitance is decreased for all these electrodes. The FPC-90 electrodes could retain 309.5 F g^−1^ (66.5%) at a current density of 20 A g^−1^. The porous structure of FPC-90 could provide effective paths for ions transport at the high current density. However, for the FPC-180, although it also shows good specific capacitance (389.6 F g^−1^) at a current density of 0.5 A g^−1^, the capacitance retention is only 48.6% when increased to 20 A g^−1^, which is much lower than FPC-90. This is owing to the difference in the morphology of Fe_2_O_3_ nanoparticles caused by excessive deposition reaction. The larger Fe_2_O_3_ loading weakens the porous characteristics and structural stability of composites, and thus reduces the specific capacitance and rate performance. In addition, despite the pure α-Fe_2_O_3_ nanorods electrode showing similar sizes to the α-Fe_2_O_3_ nanorods loaded on the surface of FPC-90, the pure α-Fe_2_O_3_ nanorods are only simple agglomerations of nanoparticles with a limited specific surface area, which could not provide enough active interfaces, resulting in poor capacitance performance. Especially when the current density is increased to higher than 10 A g^−1^, the pure α-Fe_2_O_3_ electrode could not charge–discharge effectively, which indicates that carbon substrate plays an important role in ion transport during the charge–discharge process.

Figure 4c shows the long-term cycling stability at the current density of 5 A g^−1^ for 4000 cycles. After 4000 cycles, the FPC-90 exhibits good capacitance retention of 88.4%, which is only 61.9% for FPC-180 at the same time. Firstly, it is due to the interfacial redox reaction which may cause damage to the electrode surface, thus limiting its capacitance durability. The higher Fe_2_O_3_ content in FPC-180 makes its cycling stability worse than that of FPC-90. In addition, the exfoliation of Fe_2_O_3_ nanoparticles that occurs during the long-cycling process may also reduce the cycling stability.

The electrochemical impedance characteristics of these electrodes are evaluated by the EIS test. As shown in Figure 4d, the Nyquist plots mainly include two parts: a slight semicircle in the high frequency region controlled by charge transfer and a straight line in the low frequency region controlled by ions diffusion [45]. Both FPC-90 and FPC-180 show a line nearly perpendicular to the real axis in the low frequency region, indicating that they exhibit good diffusion rates and capacitance characteristics. Based on the equivalent circuit shown as an inset in Figure 4d, the impedance data are fitted. The equivalent serials resistance (R_s_) of these electrodes follows the order of FPC-90 (1.73 Ω) < FPC-180 (2.05 Ω) < α-Fe_2_O_3_ (2.75 Ω). The R_s_ increases with the increasing of Fe_2_O_3_ loading in the FPC, due to the Fe_2_O_3_ content inhibiting the conductivity of the capacitive active materials. Additionally, it is observed that the semicircle also increases in the order of FPC-90 (1.28 Ω) < FPC-180 (1.85 Ω) < α-Fe_2_O_3_ (3.06 Ω), indicating the FPC-90 possesses the smallest interfacial charge transfer residence (R_ct_), which represents the optimal ionic diffusion capacity.

To further investigate the electrochemical capacitance mechanism of the FPC electrode, the contribution of EDLC and pseudo-capacitance on the total capacitance are evaluated by CV tests. The power law relation [46] is used as:(5)$$i=av^{b}$$
where *i* is the current response (mA) in CV curves, *v* is the scan rate (mV s^−1^), and *a* and *b* are the coefficients of the equation. The *b*-value close to 1.0 represents the ideally capacitive-controlled charge storage, and when it approaches 0.5, it indicates a diffusion-controlled process [47]. As shown in Figure 5a,b, the *b*-values could be calculated separately from the anode or cathode processes. The above equation can be easily translated into linear form, and the *b*-values could be obtained from the slope, as the inset figures illustrate. The *b*-values of FPC-90 and FPC-180 electrodes vary at around 0.8 in a large potential range, indicating that the current responses are related to both capacitive-controlled and diffusion-controlled processes. In addition, the ideal capacitive-controlled contribution and the diffusion-controlled pseudo-capacitance contribution for these electrodes are also calculated using the following formula:(6)$${i=k}_{1}{v+k}_{2}v^{1/2}\;$$
where $k_{1}v$ and $k_{2}v^{1/2}$, respectively, represent the current responses of ideal capacitive behavior and diffusion behavior [48]. Taking $k_{1}v$ as the ordinate and corresponding potential as abscissa, the EDLC is obtained by the area under the curves. In addition, the rest of the total capacitance could be regarded as diffusion-controlled capacitance. The charge storage contributions of different processes at a scan rate of 20 mV s^−1^ are shown in Figure 5c,d. The capacitance painted with a color mark shows narrow willow-leaf-like shapes for both FPC-90 and FPC-180, and the proportion of EDLC capacitance is about 43.91% and 45.02%, respectively. The charge storage contributions of these two electrodes at different scan rates are shown in Appendix A. It shows that the diffusion-controlled pseudo-capacitance contribution for FPC-90 is higher than FPC-180, even though the Fe_2_O_3_ content is lower in the former one. This also means Fe_2_O_3_ nanorods with small and uniform particles are more conducive to the pseudo-capacitance effect. The relative capacitance contributions at different scan rates are summarized in Figure 5e,f. The contribution of the EDLC process for FPC-90 is increased from 38.16% to 75.30% with the scan rates from 10 to 100 mV s^−1^, and the FPC-180 shows the same trend. At the low scan rate, the diffusion-controlled process which is attributed to the pseudo-capacitance of Fe_2_O_3_ plays a major role in charge storage, because of the sufficient diffusion time for the faradaic reaction. Conversely, at a higher scan rate, the diffusion process is inhibited, and the surface capacitance-controlled process dominates.

According to the above analysis, the role of the HPC substrate in the composite electrode could be schematically presented in Figure 6. Firstly, compared with the pure α-Fe_2_O_3_ grown on the Ti foil substrate (Figure 6a), the Fe_2_O_3_ nanorods grown on the HPC substrate exhibit a more dispersed structure (Figure 6b), which produces more interfaces between electrode and electrolyte. Hence, it improves the utilization efficiency of Fe_2_O_3_ and enhances the pseudo-capacitances. In addition, the HPC substrate allows full play to its porous advantages, which contribute to the total capacitance with the EDLC effect significantly. The hierarchical porous structure plays as a buffer during the charge storage, which facilitates ions diffusion and electron transfer. Further, the carbon substrate with high conductivity can overcome the poor conductivity of Fe_2_O_3_, which is also beneficial to the capacitance performance.

It has been known that the pseudo-capacitance effect is an interfacial phenomenon between the electrode and electrolyte. The morphology and structure of the electrode are closely related to its pseudo-capacitance performance. Here, the morphology of Fe_2_O_3_ nanorods on the FPC composite is controlled by the hydrothermal reaction temperature. As shown in Figure 7, FPC-90 maintains the relatively large specific surface area structure of the HPC substrate after fine Fe_2_O_3_ nanorods loading. This structure facilitates the ionic diffusion into the electrode materials, which means higher specific capacitance and rating performance. However, the larger size of Fe_2_O_3_ nanoparticles would block the porous structure of FPC-180. The blockage of pores and channels in FPC-180 would inevitably obstruct some electrolyte ions from entering the electrode materials to contact more active surfaces of Fe_2_O_3_ nanorods, thus limiting its capacitance performance.

To demonstrate the practical application potential of the FPC electrode, an ASC device is assembled in 1 M Na_2_SO_4_ electrolyte using FPC-90 synthesized in this work as an anode, and MnO_2_/carbon composite materials (MPC) prepared based on oily sludge in the previous work [49] as a cathode. The characterization of MPC is shown in Appendix A. As shown in Appendix A, the mass ratio of MPC and FPC loaded on cathode and anode is determined as 1:1 according to the charge balance of both electrodes. The electrochemical characteristics of the ASC device are shown in Figure 8. Figure 8a shows no obvious deformation on the CV curves, demonstrating the MPC//FPC device can maintain operation stably in the voltage of 2.0 V. Figure 8b depicts that all CV curves at different scan rates keep similar shapes, revealing the good rate performance of the device. GCD curves at different current densities in Figure 8c show a symmetrical charge and discharge time, indicating that the device possesses high coulombic efficiency. The specific capacitance of this device could be calculated according to these GCD curves. As shown in Figure 8d, the total specific capacitance is 130.1 F g^−1^ at 0.5 A g^−1^, and it still delivers 81 F g^−1^ at 20 A g^−1^. The high capacitance performance of this MPC//FPC supercapacitance device is competitive with those of MnO_2_-based or Fe_2_O_3_-based supercapacitors reported in the other literature (shown in Appendix A). The Nyquist plot of the ASC device is shown in Appendix A. Similar to the above analysis, the small semicircle and *x*-intercept in the high frequency region mean low resistance for ion diffusion and distribution, which are beneficial to the capacitance properties.

To evaluate the cycle stability of the MPC//FPC device, GCD cycling tests at 5 A g^−1^ are conducted under different voltages. A thousand cycles are performed at each voltage, and then another 1000 cycles are continued after adjusting the operating voltage. As shown in Figure 8e, the capacitance retention rates per 1000 cycles are 97.1%, 96.3%, 96.8%, 97.4%, 97.2%, and 95.3%, respectively. The slight change in capacitance loss rate indicates that the operating voltage of the device has little influence on the cycling performance. After a series of cycling tests, the device maintains 77.6% capacitance retention after 6000 cycles. Figure 8f gives the Ragone plot of this ASC device. It exhibits an energy density of up to 72.3 W h kg^−1^ at 500 W kg^−1^, which is much better than that of many Fe_2_O_3_-based ASC devices reported in the literature, such as MnO_2_/m-rGO//Fe_2_O_3_/m-rGO [50] (41.7 W h kg^−1^ at 13.5 kW kg^−1^), GF/CoMoO_4_// GF-CNT/Fe_2_O_3_ [51] (74.7 W h kg^−1^ at 1.4 kW kg^−1^), MnO_2_/CNTs// Fe_2_O_3_/CNTs [52] (45.8 W h kg^−1^ at 0.41 kW kg^−1^), 3D HPC/NiCo_2_S_4_//3DHPC/Fe_2_O_3_ [53] (44.4 W h kg^−1^ at 1.62 kW kg^−1^), et al. (more details are shown in Appendix A). Moreover, the energy density could also maintain 45 W h kg^−1^ under a high power density of 20 kW kg^−1^.

## 4. Conclusions

In this work, a composite material containing Fe_2_O_3_ nanorods and porous carbon was synthesized through a facile hydrothermal route from oily sludge waste as a potential candidate for a supercapacitor anode. The as-received FPC material possesses a unique nanostructure which is favorable to realizing the synergy of EDLC and pseudo-capacitance. The hierarchical porous carbon substrate provides plenty of active sites for EDLC and improves the utilization efficiency of Fe_2_O_3_ which enhances the pseudo-capacitance effect. The fine and uniform Fe_2_O_3_ nanorods loading is also beneficial to improve the capacitance performance. The FPC-90 electrode material could deliver a high specific capacitance of 465 F g^−1^ at 0.5 A g^−1^, favorable rate capability of 66.5% at 20 A g^−1^, and long-life cycling stability. Furthermore, the as-assembled ASC device with FPC as the anode could operate at a voltage of 0–2.0 V, which could provide an outstanding energy density of 72.3 W h kg^−1^. In addition, it also demonstrated a stable cycling performance with a capacitance retention of 77.6% after 6000 cycles. These excellent properties evidenced the merit of the Fe_2_O_3_/porous carbon composite material, which has large practical potential in energy storage applications.

## Figures and Tables

**Figure 1 nanomaterials-12-03819-f001:**
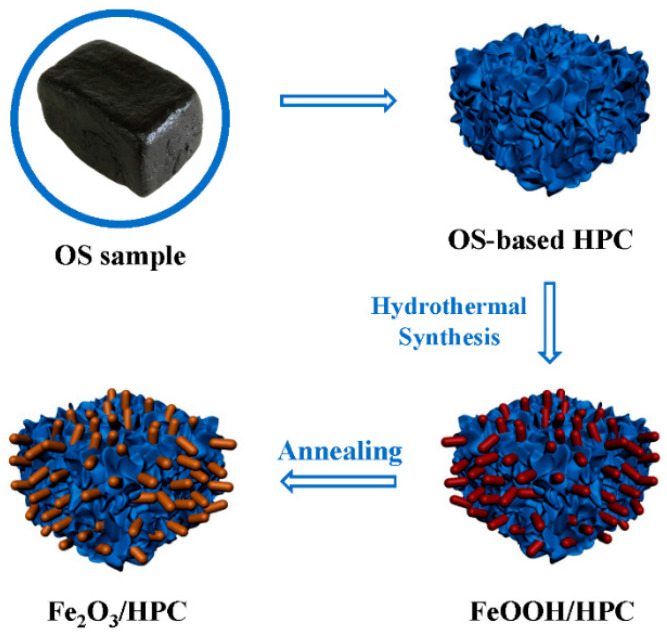
Schematic procedure for the fabrication of FPC electrode materials.

**Figure 2 nanomaterials-12-03819-f002:**
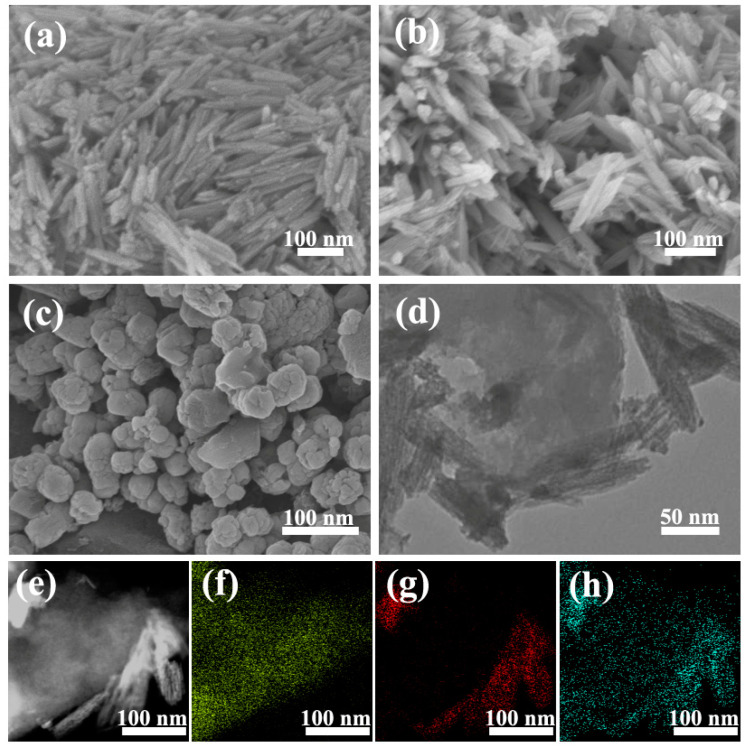
SEM images of Fe_2_O_3_ (**a**), FPC-90 (**b**), FPC-180 (**c**), and TEM images of FPC-90 (**d**,**e**) composites with corresponding elemental mapping images of C (**f**), Fe (**g**), and O (**h**).

**Figure 3 nanomaterials-12-03819-f003:**
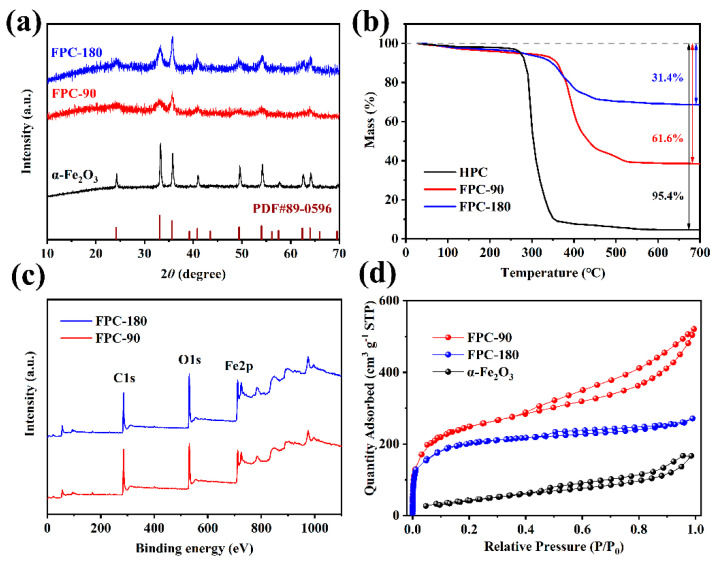
The XRD patterns (**a**), TGA analyses (**b**), XPS spectra (**c**), and N_2_ adsorption−desorption isotherms (**d**) of FPC samples.

**Figure 4 nanomaterials-12-03819-f004:**
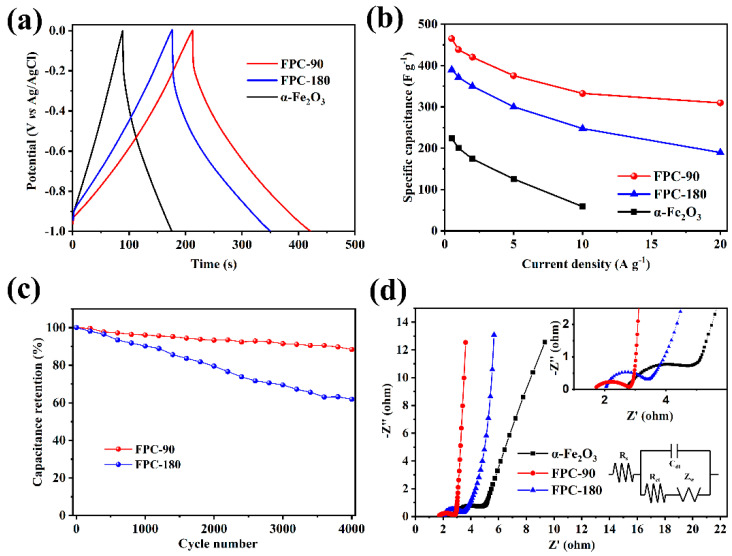
The comparison of GCD curves of different FPC samples (**a**), the rate performance of FPCs electrodes (**b**), and the long−term cycling ability of FPCs electrodes (**c**). Nyquist plot of the different electrodes: the inset is the enlarged image of the high frequency zone (**d**).

**Figure 5 nanomaterials-12-03819-f005:**
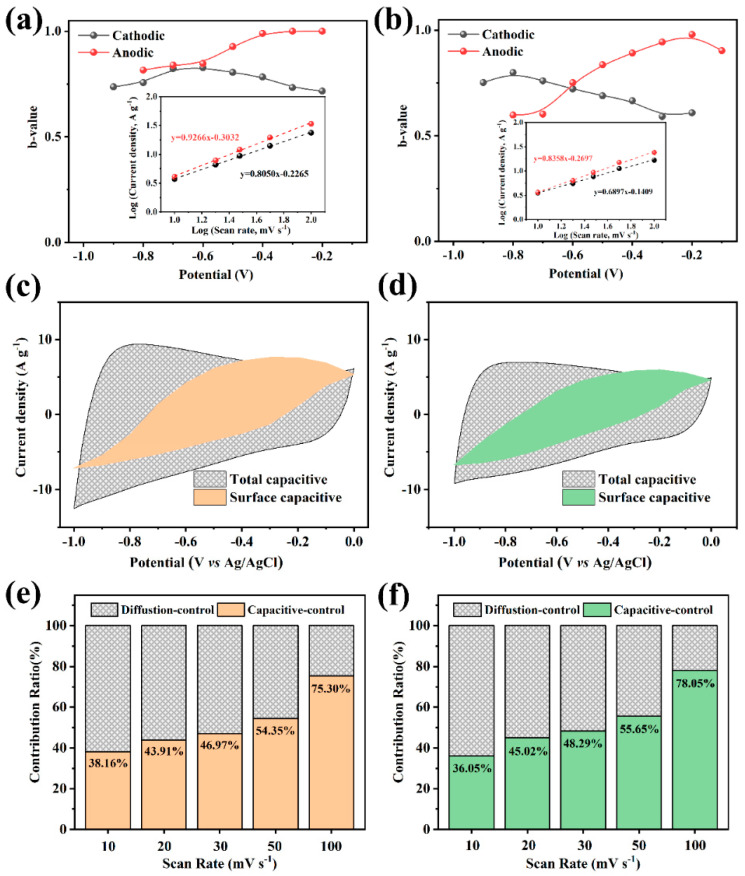
Calculated *b*-values of FPC-90 (**a**) and FPC-180 (**b**) from cathode and anode branch of CV curves, insets are the currents vs. scan rate plots of two electrodes derived from the CV curves of 20 mV s^−−1^ at a potential of 0.5 V. The charge storage contributions of FPC-90 (**c**) and FPC-180 (**d**) from the pseudo-capacitance and EDLC at a scan rate of 20 mV s^−−1^. Ratio of capacitance contributions in FPC-90 (**e**) and FPC-180 (**f**) at different scan rates.

**Figure 6 nanomaterials-12-03819-f006:**
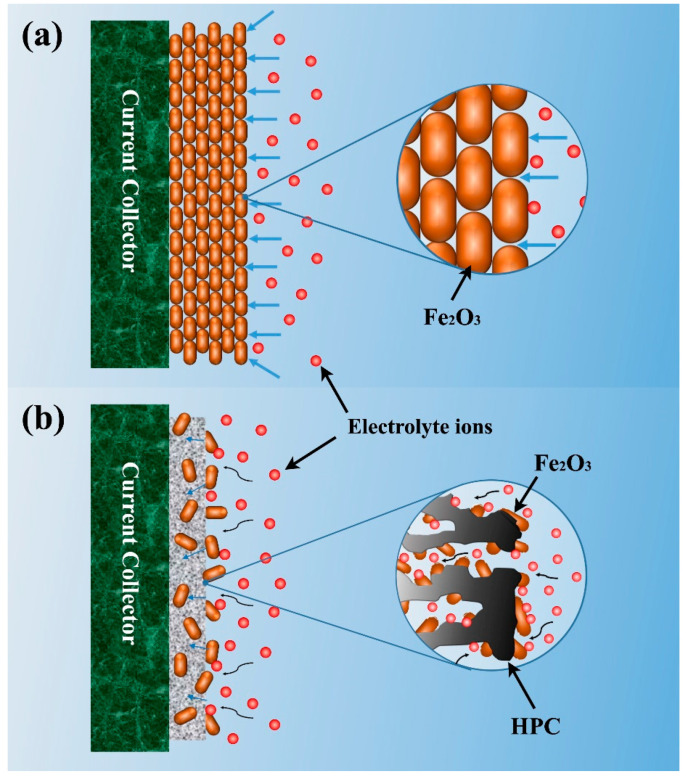
Effect of HPC substrate on the capacitance of composite electrode.

**Figure 7 nanomaterials-12-03819-f007:**
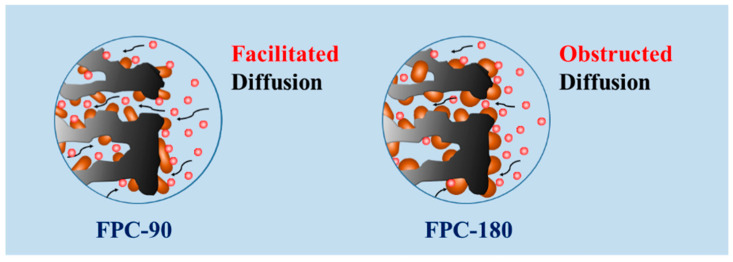
Effect of Fe_2_O_3_ on the capacitance of composite electrode.

**Figure 8 nanomaterials-12-03819-f008:**
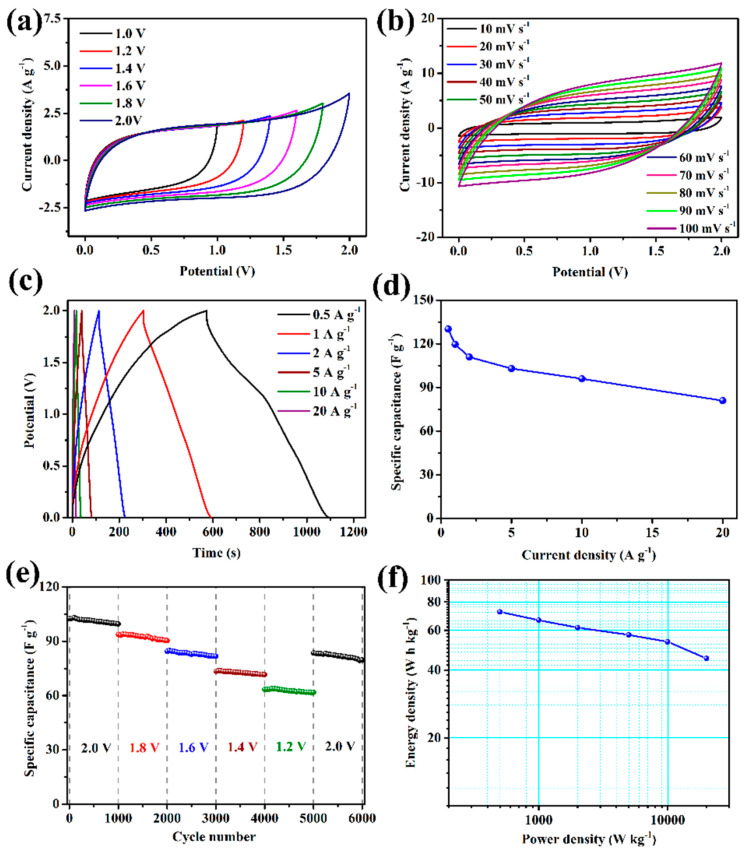
CV curves of MPC//FPC ASC device in different potential windows at a scan rate of 20 mV s^−1^ (**a**) and CV curves of the device at different scan rates (**b**). GCD curves of the device at different current densities (**c**) and specific capacitance calculated from these GCD curves (**d**). Cycling durability test of the device in different potential windows (**e**) and the Ragone plot of the device (**f**).

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
