# Peer review of "Fe2O3/Porous Carbon Composite Derived from Oily Sludge Waste as an Advanced Anode Material for Supercapacitor Application"

_nanomaterials, 2022, doi:10.3390/nano12213819_

Round 1

Reviewer 1 Report

After careful evaluation of the manuscript, I cannot recommend for publication at the current state as there are many queries need to be addressed. The reason for my decision is as follows:

Comments:

1)         The introduction section needs to be emphasized with the addition of recent articles on supercapacitors and also provide the prime novelty of the current work. Ref: Sustainable Materials and Technologies 33 (2022) e00459; Nat Electron (2022), https://doi.org/10.1038/s41928-022-00843-6

2)         The electro-active mass of the working electrode is missing in the manuscript.

3)         For the fabrication of the ASC device, the positive electrode characterization is missing in the manuscript.

4)         To fabricate the ASC, charge balance between the positive and negative electrode is must which need to be provided in the manuscript. The electroactive mass of both positive and negative electrode need to be provided in the manuscript.

5)         The Nyquist plot need to be fitted with the equivalent circuit and discuss in the manuscript. In addition, provide the Nyquist plot of the ASC device before and after stability in the revised manuscript.

Reviewer 2 Report

Supercapacitors featuring high power density, high specific energy, and excellent long-term cyclability have received great interest in many potential applications. Mapping composite materials as electrodes is an effective way to obtain electrode materials with high supercapacitive performance. Iron-based compounds are considered one of the most promising pseudocapacitor electrode materials due to their abundance, low cost, environmentally friendly, and high electrochemical performance. Authors have used hierarchical porous carbon derived from oily sludge waste on which Fe2O3 active material has been loaded. The area of research is significant, but the novelty part of the work must be emphasised in the appropriate section. Iron-based oxides are well-known electrodes (anodes) in batteries/supercapacitors.

The manuscript itself is written OK (however a few of the sentences are unclear) but the main concern of this paper and the areas that require clarification are given below.

·         In line 28, please remove the word “innovative”. Heaps of articles are available in this research area.

·         Lines 41 – 43; the second half of the sentence is unclear. “dues to the …”

·         Line 40: The authors quickly jumped to ASC even without explaining it.

·         “Especially for ASC, the research on the cathode and anode are in the same position” no reader can understand this sentence.

·         Line 46 – 47: lots of research efforts have been devoted but only two references have been shown. Either provide more references or tweak the sentence. Either way, key papers by M. Minakshi et al. on-cathode materials for high-performance supercapacitors can be included.

·         Line 53; binary metal oxides such as MgMoO4, NiMoO4, CoMoO4, and Co3O4 have also been widely used as anodes for supercapacitors. Please elaborate.

·         Fe2O3 is an insulator (with a low order of conductivity) at room temperature while Fe3O4 possesses metallic electrical conductivity (two orders of magnitude higher). Being this is the case, why Fe2O3 has been chosen?

·         Due to its huge surface area, massive electrical conductivity, high mechanical power, and less density, traditionally activated carbon (or modified version of this material with a higher porosity) have been commonly employed as anode materials for supercapacitors.  

·         Line 73: where/how the oily sludge waste has been produced?

·         Line 77; please mention the electrolyte (aqueous/non-aqueous)?

·         Line 95: what is HPC-A?

·         What is OS in figure 1? Oily sludge needs to be labeled as OS if this reviewer is correct.

·         Section 3: Does the role of carbon in the anode facilitate the volume expansion/contraction, or otherwise assist the electron transfer during charge storage process that eventually improves the performance characteristics?

·         How much is Fe2O3 loading on HPC?

·         Why are CV curves not shown in Figure 4?

·         The obtained ASC capacitance In Figures 5 and 8 can be compared with the similar metal oxides (MnO2) recently reported in the literature such as doi.org/10.1016/j.ceramint.2022.03.266; and doi.org/10.1016/j.est.2022.105403.

·         What can one infer from Fig. 4a; pseudocapacitive faradaic characteristic?

·         Th effect of combining HPC and pseudocapacitive Fe2O3 and its potential for energy storage discussion need to be bit more critical.

·         How MPC has been synthesized? What is the rationale for choosing MnO2?

Reviewer 3 Report

The proposed manuscript describes the preparation of a composite material containing Fe2O3 nanorods and porous carbon. The obtained material is investigated from the point of view of its application as an anode of a pseudocapacitor. Its advantages are demonstrated, characteristics are estimated from a wide range of experimental techniques. Simple mechanisms of functioning are considered. The obtained properties testify to the practical importance of the investigated Fe2O3/pore-containing carbon composite material for energy storage devices.

Some references relevant to the research should be added

https://doi.org/10.1016/j.matchemphys.2019.05.047

https://doi.org/10.3390/nano12050739

I think that the article can be accepted after correcting minor typos and improving some of the wording, for example, difficult and easy diffusion are not common terms. Probably, obstructed and facilitated diffusion are more suitable terms.

Round 2

Reviewer 1 Report

I recommend for acceptance at the current state.

Reviewer 2 Report

I have read the revised part of the manuscript and the author's responses to my queries. It appears the quality of the manuscript has been improved. To my opinion, it is OK to publish.